# Epileptic Patients with More Clinic Visits Are More Likely to Be Diagnosed with Dementia—A Population-Based Retrospective Cohort Study

**DOI:** 10.3390/diagnostics14232748

**Published:** 2024-12-06

**Authors:** Pao-Sheng Yen, Chih-Hsin Muo, Chung-Hsin Yeh, Fung-Chang Sung

**Affiliations:** 1Department of Neuroradiology, Kuang Tien General Hospital, Taichun 43303, Taiwan; 5322@ktgh.com.tw; 2Department of Nursing, Hungkuang University, Taichung 43303, Taiwan; 3Management Office for Health Data, China Medical University Hospital, Taichung 40447, Taiwan; b8507006@gmail.com; 4Department of Neurology, Yuan Rung Hospital, Changhua 51045, Taiwan; 5Department of Nursing, College of Nursing and Health, Da-Yeh University, Changhua 51591, Taiwan; 6Department of Health Services Administration, College of Public Health, China Medical University, Taichung 40402, Taiwan; 7Department of Food Nutrition and Health Biotechnology, Asia University, Taichung 41354, Taiwan

**Keywords:** epilepsy, epilepsy treatment gap, retrospective cohort study, dementia, compliance

## Abstract

Objective: This retrospective cohort study assessed dementia risk in epilepsy patients associated with the compliance to epileptic treatment visits. Methods: We used Taiwanese insurance claims data to establish an epilepsy cohort (*N* = 39,216) diagnosed in 2000–2015 and a matched control cohort without epilepsy (*N* = 156,864), evaluating the incident dementia by the end of 2016. Results: The dementia incidence was 2.9-fold higher in the epilepsy cohort than in comparisons (4.68 vs. 1.59 per 1000 person-years). Only 9.3% of epilepsy patients were compliant to ≥80% of scheduled treatment visits, but they exhibited a 7.2-fold higher dementia incidence than those without treatment. The contrast was greater in younger patients than in the elderly (20-fold versus 5.5-fold). Dementia incidence increased with the frequency of neurological consultations, peaking in the first year after epilepsy diagnosis. Conclusions: Epileptic patients with more clinical visits for active treatment had a higher chance of dementia diagnosis, highlighting the importance of close neurological monitoring post-epilepsy diagnosis to address potential dementia complications.

## 1. Introduction

Epilepsy is a highly prevalent chronic brain disorder, affecting 70 million individuals globally and over 3 million in the United States [1,2]. It manifests as repetitive seizures or seizure clusters, temporary episodes of involuntary movement that can lead to neurological, cognitive, and psychosocial consequences [2,3,4,5]. Alarmingly, nearly 80% of epilepsy patients reside in low- and middle-income countries, where appropriate diagnosis and treatment are often lacking [2]. However, up to 70% of individuals with epilepsy can be seizure free if they received suitable care [6].

Addressing the epilepsy treatment gap (ETG) is a pressing global challenge [2]. A retrospective cohort study in the United States discovered that 36.7% of newly diagnosed individuals with epilepsy remained untreated for up to three years after diagnosis [7]. Recent reports have highlighted the association between ETG and socioeconomic status in the United States [8]. In the Philippines, the gap is influenced not only by socioeconomic limitations but also by a lack of public support [9]. The elderly citizens with epilepsy faces a more significant challenge due to ETG [10].

Misunderstandings and social stigma surrounding epilepsy result in many affected individuals avoiding treatment, leading to a significant treatment gap known as ETG. Shockingly, research indicates that over 75% of epilepsy patients remain untreated, greatly impacting their health and well-being [4,5]. These untreated individuals face an elevated risk of injuries, hospitalization, and even suicide, resulting in increased healthcare costs and resource allocation compared to those who receive adequate care [7,11]. Furthermore, uncontrolled epilepsy significantly affects cognitive performance, placing patients at a higher vulnerability [12,13].

A previous three-year study conducted in Taiwan following 100 patients with cryptogenic epilepsy. Utilizing a cognitive ability screening instrument (CASI), the study revealed that 36% of patients developed cognitive impairment [14]. This deterioration was associated with factors such as education level, seizure types, duration of epilepsy, and the use of multiple therapies [14,15]. The finding underscores the potential benefits of appropriate treatment in mitigating the negative impact of seizures on cognitive function.

Research has shown an elevated risk of developing dementia among individuals with epilepsy. A prospective case-control study found a more than 7-fold higher risk of psychiatric disorders, including dementia, in adults with active epilepsy and intellectual disability [16]. In the recent longitudinal, multicenter cohort studies, Zawar et al. found that both cognitively normal adults and those with mild cognitive impairment with poor-controlled/active seizures were at a near 2-fold higher risk to develop earlier cognitive decline [17,18]. Evidence suggests a bidirectional relationship between dementia and epilepsy [13]. Furthermore, recent research indicates shared pathogenesis between dementia, Alzheimer’s disease, and late-onset epilepsy [19,20].

Depression, dementia, psychosis, and epilepsy may all be classified as disorders of brain networks [21,22]. Moreover, clinic visits for appropriate treatment may significantly reduce adverse cognitive effects associated with epilepsy. In light of this, a population-based longitudinal study was conducted to explore whether individuals with epilepsy who have more frequent appointments for clinic visits for their treatment regimen have a lower risk of developing dementia. This study analyzed population data from the Taiwan’s National Health Insurance Research Database, which provides coverage to over 99% of the country’s residents [23].

By investigating the relationship between clinic visit frequency and the occurrence of dementia in individuals with epilepsy, this study aimed to shed light on the potential benefits of treatment compliance, particularly for the elderly patients. The findings from this study could enhance our understanding of the importance of treatment compliance, not only for seizure control but also for reducing the risk of cognitive decline and dementia in individuals with epilepsy.

## 2. Methods

### 2.1. Data Source

This study used the real-world data of the National Health Insurance Research Database and death registry of Taiwan from 2000 to 2016. The insurance database tally medical claims data for outpatient and inpatient services in Taiwan provide information on treatments, prescriptions, and costs for services. Identifications of the insured population in the database were re-coded with surrogate identifications for privacy protection [23]. In the database, drug and disease classifications conformed to the Anatomical Therapeutic Chemical (ATC) Classification System and the International Statistical Classification of Diseases and Related Health Problems, 9th and 10th Revisions (ICD-9 and ICD-10). This study was approved by the Research Ethics Committee of China Medical University and Hospital (CRREC-107-021).

### 2.2. Study Subjects

We identified patients who had been diagnosed with seizures (or epilepsy) at least twice for the potential epileptic cohort. From the whole insured population, 320,559 cases of epilepsy were diagnosed in 2000–2015 (Figure 1). After excluding 281,343 patients who were ineligible for this study, 39,216 cases of epilepsy patients were included in the epilepsy cohort. The date epilepsy was diagnosed was defined as the index date. We used a similar method to exclude 1,620,117 persons among individuals without any history of epilepsy. From 8,917,940 persons without a history of epilepsy and/or dementia, the comparison cohort were selected and matched by propensity scores with a sample size four time that of the epilepsy cohort. Individuals in both cohorts were followed until dementia was diagnosed, death occurred, withdrawal from the insurance occurred, or the end of 2016 arrived. The follow-up person-years were measured for each person.

### 2.3. Comorbidity, Anti-Epilepsy Compliance Rate, and Neurologic Visit Frequency

We also considered comorbidities of hypertension, diabetes, hyperlipidemia, coronary heart disease, atrial fibrillation, chronic obstructive pulmonary disease (COPD), cancer, heart failure, depression, liver disease, and chronic infection/inflammation as covariates in this study. The comorbidity that had been diagnosed in at least three outpatient visits, or once in inpatient care at baseline, was defined for inclusion. In order to evaluate whether a patient who adhered to the epilepsy treatment was associated with the diagnosis of dementia, we calculated rates of taking anti-epilepsy drugs (AEDs) by the sum of days taking AEDs in the follow-up days from the index date to the end of follow-up for each individual [24]. We divided those patients into 3 groups: taking AEDs for >800 days, 400–799 days, and 1–399 days per 1000 follow-up days as compliance rates of ≥80%, 40–79%, and 1–39%, respectively. Those without the medication were in the non-compliant group of 0%. The compliance rate reflects how active a patient was in taking the prescribed medication. Furthermore, we also evaluated the dementia diagnosis associated with neurological consultation frequency in the epileptic cohort. 

### 2.4. Statistical Analysis

We compared distributions of sex, age, income, urbanization level, and comorbidities between epilepsy and comparison cohorts. Balances of demographics and comorbidities between the two cohorts were presented by standardization differences, with the standardization difference over 0.1 being defined as unbalanced. The cumulative incidence of dementia in two cohorts was calculated and plotted by the Kaplan–Meier method and examined by the log-rank test. We calculated the incidence rate of dementia for each cohort. For the epilepsy cohort, we further calculated the dementia incidence by compliance rate, and the patient compliance to treatment regimen: ≥80%, 40–79%, 1–39%, and 0%. The epilepsy cohort to the comparison cohort hazard ratio (HR) and related 95% confidence interval (CI) were assessed using Cox proportional hazards regression. The adjusted hazard ratio (aHR) was calculated after controlling for matched pairs. The HRs were also estimated by sex and compliance rate, age and compliance rate, and by follow-up years (<1.0, 1.0–2.9, 3.0–4.9 and ≥5.0 years). We further conducted a nested case-control analysis comparing dementia cases and non-dementia patients for the epilepsy cohort, attempting to identify whether demographic status, comorbidities, and neurologic consultation frequency were associated with the diagnosis of dementia in the epilepsy cohort. Odds ratios (ORs) of dementia with 95% CIs were estimated using logistic regression analysis. All statistical tests were two-sided, and the statistical significance was defined as *p*-value < 0.05. We used SAS, version 9.4 (SAS Institute, Cary, NC, USA).

## 3. Results

### 3.1. Baseline Characteristics of Study Cohorts

Table 1 shows that distributions of demographic status and most comorbidities were similar in the epilepsy cohort and the comparison cohort, with nearly 53% being men and 19% being the elderly. Prevalence rates of atrial fibrillation and heart failure at baseline were slightly higher in the epilepsy cohort than in comparisons, but significant.

### 3.2. Incidence of Dementia Between Cohorts by Sex and Compliance Rate

During the follow-up, the cumulative incidence of dementia was about 4% higher in the epilepsy cohort than in the comparisons, and it was similar for men and women (log-rank test *p* < 0.0001) (Figure 2). Table 2 shows the incidence of dementia was nearly three times higher in the epilepsy cohort than in comparisons (4.68 versus 1.59 per 1000 person-years), with an aHR of 2.92 (95% CI = 2.70–3.13). In the epileptic cohort, patients who were active in adhering to clinic visits for over 80% of days had a much higher rate of being diagnosed with dementia, with an incidence of 33.9 per 1000 person-years or an aHR of 17.3 (95% CI = 14.4–2.07). The incidence rate decreased with the decreasing compliance rate. Near 30% of patients were untreated with an incidence of 4.68 per 1000 person-years with an aHR of 2.91 (95% CI = 2.58–3.29). Both men and women had similar trends.

### 3.3. Incidence of Dementia by Age and Compliance Rate

Table 3 shows that dementia incidence increased with age in both cohorts. In the epilepsy cohort, the incidence increased from 1.10 per 1000 person-years in the younger group to 26.9 per 1000 person-years in the elderly. Among those with a compliance rate of ≥80%, the incidence was nearly 13 times higher in the elderly than in the young (169.2 versus 13.2 per 1000 person-years). However, the aHR was, relatively, much greater for the younger epilepsy patients than the elderly ones (205 vs. 12.3). Among patients with an 80% compliance rate, younger patients were more likely than the elderly to be diagnosed with dementia (20-fold versus 5.5-fold), compared to corresponding patients with a 0% compliance rate.

### 3.4. Incidence of Dementia by Follow-Up Years

Among 1035 cases with dementia developed in the epileptic cohort, 398 cases (38.5%) occurred within the first year of follow-up (aHR = 6.06, 95% CI = 5.24–7.00) (Table 4). The incidence decreased with the follow-up year, with an aHR that declined to 1.93 (95% CI = 1.68–2.21) after 5.0 years or longer of follow-up.

### 3.5. Nested Case-Control Analysis of Dementia in Epilepsy Cohort

The nested case-control analysis shows that most of comorbidities were more prevalent in dementia cases than in non-cases (Table 5), but no comorbidities could explain the risk of developing dementia. Female epilepsy patients had an aOR of 1.32 (95% CI = 1.16–1.50) to develop dementia compared with male patients. Patients diagnosed with dementia increased with neurological consultations. Epilepsy patients who had >20 neurological visits were 2.6 times more likely to be diagnosed with dementia than those without the visit, with an aOR of 4.04 (95% CI = 3.15–5.18).

## 4. Discussion

### 4.1. Epilepsy Risk, Treatment Compliance, and Socio-Demographics

The risks of developing epilepsy and subsequent dementia vary among populations [11,16,19,20,25,26]. In our real-world data, we observed that 1.24% of the population had epilepsy (Figure 1), which might be higher than the annual prevalence of 1.1% reported in the United States [25,26].

It has been reported that populations in low- and middle-income countries are at a higher risk of epilepsy associated with weak healthcare systems, poor hygiene, and insufficient knowledge, tending to affect the elderly [2]. In our study, we identified a higher proportion of men than women with epilepsy, and only 11.9% of epilepsy cases in the least urbanized area. Our elderly patients, accounting for only 19% of all patients, were more likely to reside in rural areas. We suspect that the rapid aging of Taiwan’s population may explain this finding [27,28]. However, they had no challenges in accessing healthcare.

Known as National Health Insurance, and instituted in 1995 by merging all public health insurance programs, a universal healthcare system covers all residents in Taiwan. This healthcare provides high-quality comprehensive care coverage, with a low burden of healthcare costs [23]. However, we were surprised to find that only 9.3% of patients with epilepsy in our study adequately adhered to their treatment, defined as a compliance rate of over 80%. They were much more likely to be diagnosed with dementia. This indicates a significant gap of AEDs among epilepsy patients in Taiwan. Notably, among the elderly, the rate of good treatment compliance was higher compared to younger individuals (15.9% [1176 out of 7404] versus 6.6% [1025 out of 15,529]).

### 4.2. Treatment Compliance and Dementia Detection

The availability of epilepsy treatment [ETGs] poses a significant challenge worldwide, with varying rates of ETG ranging from 5.6% in Norway to 100% in some developing countries [29]. The barriers hinder the accessibility of healthcare services, including essential pharmacotherapy and surgical options for epilepsy patients [9,30,31]. Previous research has consistently demonstrated that epilepsy patients who receive adequate treatment experience show an improved quality of life and reduced seizure activity [5,11]. On the other hand, patients with poorly controlled epilepsy may face various challenges such as low self-esteem and psychiatric symptoms including anxiety, depression, and the risk of sudden unexpected death associated with epilepsy (SUDEP) [5]. A recent study from Milan found that AEDs can help reduce adverse cognitive effects in epilepsy patients [12]. In our study, we hypothesized a lowered dementia risk as the benefit of good compliance with making a clinic appointment for epilepsy treatment, particularly among the elderly population. To our surprise, we discovered a proportional relationship between compliance rate and dementia incidence, which increased with age. The incidence of dementia in patients with a treatment compliance of 80% or higher was 7.2 times higher compared to those who did not receive medication (33.9 vs. 4.68 per 1000 person-years). The incidence was nearly 13 times higher in those aged over 65 years compared to those aged 30–44 years old. This suggests that patients with better treatment compliance have a higher likelihood of being diagnosed with dementia, with a more pronounced impact observed in the elderly. To our knowledge, there are no previous studies that have reported such astounding findings. We suspect that patients with more clinic visits for AEDs may actually have a higher frequency of severe seizures, leading to more frequent clinic visits, increased medical attention, and the subsequent diagnosis of dementia.

### 4.3. The Role of Neurological Consultation

The nested case-control analysis conducted within the epilepsy cohort provided further insights into the puzzle. The results revealed that the odds of being diagnosed with dementia increased with the number of neurological clinic visits.

An earlier study in Taiwan reported that nearly half of epilepsy patients chose traditional Chinese medicine and temple worship as treatment options [32]. A recent study on medical care-seeking behaviors in rural communities of Taiwan found that the elderly population is less aware of disease severity, leading to delayed treatment [33]. However, this disadvantage did not seem to apply to the elderly patients in our study. Approximately 48% of patients had more than 20 visits for neurological consultations, indicating higher treatment compliance. On the other hand, around 8.0% of patients had no consultations, suggesting lower treatment compliance. Elderly epilepsy patients were more likely to have frequent neurological consultations, indicating a higher treatment compliance rate. Epilepsy patients without medication and with treatment compliance rates below 40% might have delayed medical treatment, making it less likely for dementia to be diagnosed.

Our case-control analysis revealed that epilepsy patients living in the least urbanized rural areas were at an elevated risk of dementia. High insurance coverage, easy accessibility, and a low copayment of the healthcare system in Taiwan lead to highly frequent clinical visits [34]. Individuals living in areas with the lowest urbanization levels have the least chance of using medical services compared to those in areas with higher urbanization levels. The medical divergence between urbanization levels are a major factor associated with the possibility and frequency of medical services [35,36,37]. We suspect that epilepsy patients living in rural areas are likely to have fewer clinic visits. The elevated dementia risk in rural areas was likely true, which might be associated with poor rural risk factors instead of treatment compliance. The recent studies reported that patients with poor compliance or fewer neurological consultations for seizures might display quicker decay in cognitive functions based on recent research [17,18].

Current healthcare institutions in Taiwan are divided into four levels: medical centers, regional hospitals, district hospitals, and clinics. Obviously, large hospitals located in higher urbanization areas provide higher healthcare-seeking choices and behaviors due to more diversity in medical care types with easy accessibility and a low medical cost burden. The “hierarchical medical system” fails to reduce unnecessary healthcare-seeking and medical resource wastage [35,36].

While seizures are common in epilepsy patients, it is surprising to observe that 38.5% (398/1035) of dementia cases were diagnosed within one year after the diagnosis of epilepsy. The incidence of dementia decreased from 12.4 per 1000 person-years in the first year to 2.99 per 1000 person-years after five years of follow-up. Patients with intensive neurological consultations may have had their dementia diagnosed earlier. Previous studies have documented a bidirectional relationship between epilepsy and dementia [13,38,39]. A recent study using data from the Framingham Heart Study found that dementia patients had a hazard ratio (HR) of 1.82 for developing epilepsy, while epilepsy patients had a HR of 1.99 for dementia [40]. Our data support a nearly 6-fold increased risk of developing dementia soon after the diagnosis of epilepsy. This extraordinary finding has not been previously reported.

## 5. Conclusions

Our findings underscore the importance of the nature of complexity in treatment adherence and ETGs for epilepsy patients. Patients with dementia are, indeed, at a higher risk of experiencing subsequent disorders and a decline in quality of life. In this study, we observed that patients with epilepsy had a nearly 3-fold increased risk of developing dementia. However, only a small proportion of epilepsy patients demonstrated good compliance to medication therapy, but they were more likely to be diagnosed with dementia. This finding suggests that patients with more frequent clinic visits may experience more recurrent seizures and frequent neurological consultations, which, in turn, increases their chances of being diagnosed with dementia. It highlights the importance of healthcare providers to understanding the complex relationship between treatment compliance, recurrent seizures, and the risk of dementia. It is essential to closely monitor patients with epilepsy after diagnosis to ensure they receive adequate care and appropriate interventions. It is crucial for both patients and healthcare providers. Further research is needed to gain deeper insights into the underlying factors contributing to these observations, and to develop strategies that ensure optimal treatment and care for epilepsy patients.

## 6. Limitations

This study has the advantage of utilizing a large population dataset to conduct a propensity score matching retrospective cohort study to assess the risk of dementia in patients with epilepsy. However, it also has some limitations that should be taken into consideration. First, approximately 79% of epileptic patients in this study had fewer clinic visits with compliance rates of <40%. This suggests that these patients may have been reluctant to seek treatment, which could potentially impact the findings. Conversely, only 9% of patients with high compliance rates (>80%) had received aggressive treatment. This could be associated with an increased diagnosis of epilepsy-related disorders, including dementia. The lack of information on the severity of epilepsy in the insurance claims data limits the ability to assess the impact of severity on the development of dementia.

Second, certain risk factors associated with the development of dementia, such as blood pressure, blood sugar levels, lipoprotein levels, and lifestyle factors, were not available for assessment in the dataset. This limits the ability to fully account for these factors in the analysis. However, the nested case-control analysis did not find a significant association between comorbidities and the occurrence of dementia.

Third, data on cognitive impairment levels, measured by tools such as CASI, Mini-Mental State Examination (MMSE), and Clinical Dementia Rating (CDR), were not available. Therefore, the assessment of the level of cognitive impairment in the study population was not possible.

Lastly, the dataset did not include information on the use of over-the-counter or complementary and alternative medicines, which could be relevant to the study. The impact of poor seizure control on the risk of developing dementia could not be evaluated due to the lack of data on patients with poor controlled status.

These limitations should be acknowledged when interpreting the findings of the study. Further research addressing these limitations could provide a more comprehensive understanding of the relationship between epilepsy, treatment compliance, and the risk of dementia.

## Figures and Tables

**Figure 1 diagnostics-14-02748-f001:**
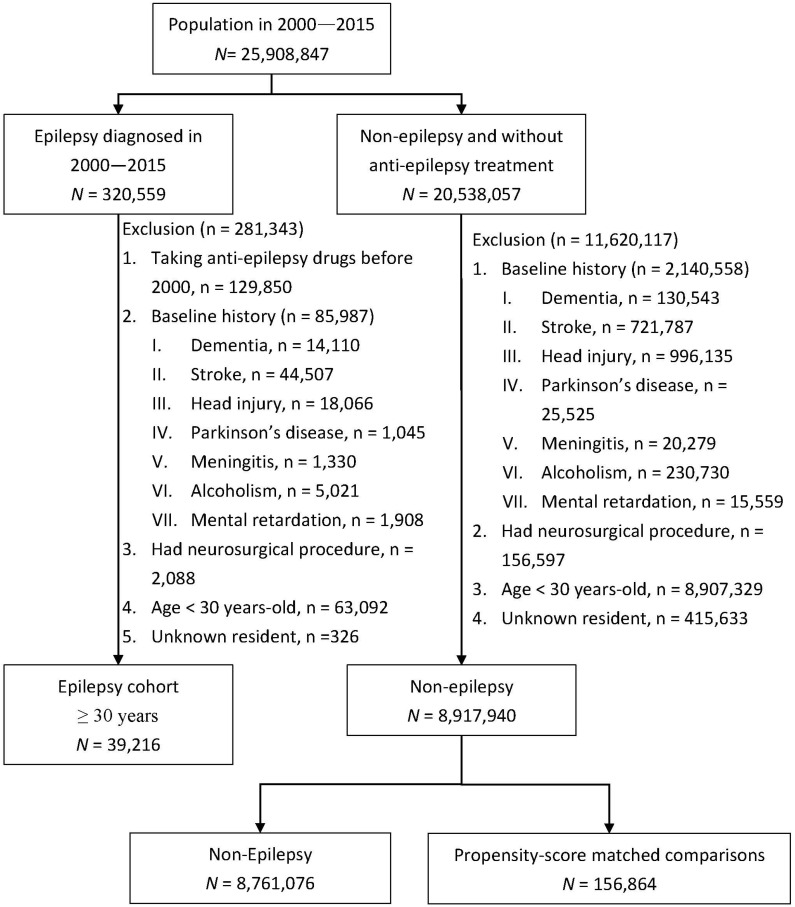
Flow chart for establishing study cohorts with and without epilepsy.

**Figure 2 diagnostics-14-02748-f002:**
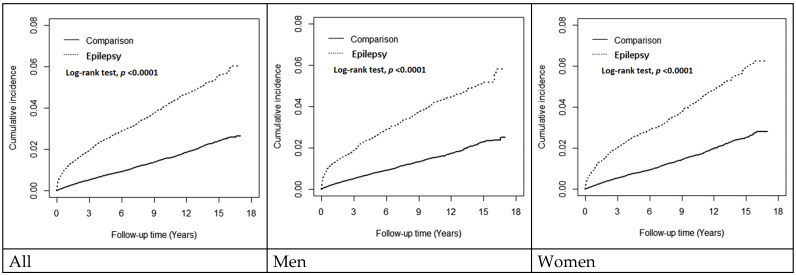
Kaplan–Meier plot for cumulative incidence of dementia in epilepsy cohort and comparisons.

**Table 1 diagnostics-14-02748-t001:** Baseline characteristics compared between cohorts with and without epilepsy.

	Epilepsy *N* = 39,216		Comparisons *N* = 156,864		Standardization Difference
	n	%	n	%	
Sex					
Male	20,835	53.1	83,737	53.4	0.005
Female	18,381	46.9	73,127	46.6	0.005
Age, years					
30–44	15,529	39.6	61,356	39.1	0.010
45–64	16,283	41.5	65,664	41.9	0.007
65+	7404	18.9	29,844	19.0	0.004
Mean (SD)	50.8	(14.5)	50.8	(14.4)	0.006
Income, NTD					
<19,200	12,828	32.7	51,428	32.8	0.002
19,200–28,799	16,160	41.2	64,910	41.4	0.003
28,800+	10,228	26.1	40,526	25.8	0.006
Residential Urbanization					
1 (highest)	10,531	26.9	41,887	26.7	0.003
2	11,760	30.0	47,178	30.1	0.002
3	12,256	31.3	49,174	31.4	0.002
4 (lowest)	4669	11.9	18,625	11.9	0.001
Comorbidity					
Hypertension	11,451	29.2	46,466	29.6	0.009
Diabetes	5726	14.6	22,972	14.6	0.001
Hyperlipidemia	7118	18.2	28,739	18.3	0.004
Coronary heart disease	5301	13.5	21,129	13.5	0.001
Atrial fibrillation	500	1.27	1425	0.91	0.035
COPD	5874	15.0	22,478	15.0	0.000
Cancer	11,038	28.2	44,756	28.5	0.009
Heart failure	1454	3.71	4792	3.05	0.036
Depression	3022	7.71	12,058	7.69	0.001
Liver disease	7804	19.9	31,653	20.2	0.007
Chronic infection	1068	2.72	3680	2.35	0.024

COPD, chronic obstructive pulmonary disease.

**Table 2 diagnostics-14-02748-t002:** Incidence of dementia and related epilepsy cohort to comparison cohort hazard ratio by sex and compliance rate of clinic visit.

					Hazard Ratio (95% CI)	
Compliance Rate	*N*	Event	PY	Rate	Crude	Adjusted
All						
Comparisons	156,864	1713	1,077,085	1.59	1.0	1.0
Epilepsy cohort	39,216	1035	221,379	4.68	2.91 (2.69–3.14) ***	2.92 (2.70–3.13) ***
≥80%	3649	134	3950	33.9	17.3 (14.4–20.6) ***	17.3 (14.4–2.07) ***
40–79%	4534	174	19,216	9.05	5.45 (4.67–6.38) ***	5.45 (4.67–6.37) ***
1–39%	19,634	433	135,417	3.20	2.01 (1.81–2.24) ***	2.01 (1.81–2.24) ***
0%	11,399	294	62,796	4.68	2.91 (2.57–3.29) ***	2.91 (2.58–3.29) ***
Men						
Comparisons	83,737	850	556,430	1.53	1.0	1.0
Epilepsy cohort	20,835	506	108,651	4.66	3.00 (2.68–3.34) ***	3.00 (2.69–3.34) ***
≥80%	2119	75	2106	35.61	17.9 (14.1–22.7) ***	17.9 (14.0–22.8) ***
40–79%	2508	85	9813	8.66	5.36 (4.29–6.70) ***	5.36 (4.29–6.70) ***
1–39%	10,414	194	67,423	2.88	1.88 (1.61–2.20) ***	1.88 (1.61–2.20) ***
0%	5794	152	29,309	5.19	3.34 (2.81–3.97) ***	3.34 (2.82–3.96) ***
Women						
Comparisons	73,127	863	520,655	1.66	1.0	1.0
Epilepsy cohort	18,381	529	112,728	4.69	2.81 (2.52–3.13) ***	2.81 (2.53–3.12) ***
≥80%	1530	59	1844	31.99	16.5 (12.7–21.5) ***	16.5 (12.6–21.7) ***
40–79%	2026	89	9403	9.47	5.54 (4.45–6.89) ***	5.54 (4.45–6.88) ***
1–39%	9220	239	67,994	3.52	2.12 (1.84–2.45) ***	2.12 (1.84–2.45) ***
0%	5605	142	33,487	4.24	2.53 (2.12–3.03) ***	2.53 (2.13–3.01) ***

PY, person-years; rate, per 1000 person-years; CI, confidence interval. *** *p* < 0.001.

**Table 3 diagnostics-14-02748-t003:** Incidence of dementia and related epilepsy cohort to comparison cohort hazard ratio by age and compliance rate of clinic visits.

					Hazard Ratio (95% CI)	
Compliance Rate	*N*	Event	PY	Rate (‰)	Crude	Adjusted
Age, 30–44 years						
Comparisons	61,356	26	499,203	0.05	1.0	1.0
Epilepsy cohort	15,529	120	108,957	1.10	20.8 (13.6–31.8) ***	20.8 (13.6–31.8) ***
≥80%	1025	27	2052	13.2	205 (119–354) ***	205 (121–349) ***
40–79%	1867	28	11,047	2.53	46.7 (27.4–79.6) ***	46.7 (27.4–79.4) ***
1–39%	8248	45	65,617	0.69	13.1 (8.10–21.3) ***	13.1 (8.10–21.3) ***
0%	4389	20	30,242	0.66	12.5 (6.98–22.4) ***	12.5 (6.98–22.4) ***
Age, 45–64 years						
Comparisons	65,664	223	443,336	0.50	1.0	1.0
Epilepsy cohort	16,283	302	89,596	3.37	6.47 (5.44–7.69) ***	6.47 (5.43–7.69) ***
≥80%	1448	27	1425	18.9	36.2 (24.1–54.3) ***	36.2 (24.0–54.6) ***
40–79%	1811	64	6792	9.42	18.5 (14.0–24.4) ***	18.5 (14.0–24.4) ***
1–39%	8204	141	55,443	2.54	4.86 (3.93–6.00) ***	4.86 (3.92–6.02) ***
0%	4820	70	25,936	2.70	5.20 (3.97–6.80) ***	5.20 (3.98–6.78) ***
Age, 65+ years						
Comparisons	29,844	1464	134,547	10.9	1.0	1.0
Epilepsy cohort	7404	613	22,826	26.9	2.40 (2.19–2.64) ***	2.40 (2.19–2.64) ***
≥80%	1176	80	473	169.2	12.3 (9.76–15.5) ***	12.3 (9.58–15.8) ***
40–79%	856	82	1377	59.5	5.18 (4.14–6.47) ***	5.18 (4.09–6.55) ***
1–39%	3182	247	14,357	17.2	1.56 (1.36–1.78) ***	1.56 (1.36–1.78) ***
0%	2190	204	6618	30.8	2.78 (2.40–3.22) ***	2.78 (2.39–3.23) ***

PY, person-years; rate, per 1000 person-years; CI, confidence interval. *** *p* < 0.001.

**Table 4 diagnostics-14-02748-t004:** Incidence and related epileptic cohort to comparison cohort hazard ratio of dementia by follow-up year.

	Epilepsy Cohort *N* = 39,216			Comparison Cohort *N* = 156,864			Hazard Ratio (95% CI)	
Follow-Up	Event	PY	Rate	Event	PY	Rate	Crude	Adjusted
<1.0, year	398	32,004	12.4	304	151,721	2.00	6.06 (5.22–7.04) ***	6.06 (5.24–7.00) ***
1.0–2.9	204	51,753	3.94	412	263,511	1.56	2.52 (2.13–2.98) ***	2.52 (2.13–2.98) ***
3.0–4.9	142	40,233	3.53	298	207,483	1.44	2.46 (2.01–3.00) ***	2.46 (2.01–3.00) ***
≥5.0	291	97,388	2.99	699	454,370	1.54	1.93 (1.68–2.21) ***	1.93 (1.68–2.21) ***

PY, person-years; Rate, per 1000 person-years; CI, confidence interval. *** *p* <0.001.

**Table 5 diagnostics-14-02748-t005:** Nested case-control analysis of dementia in epilepsy cohort.

	Dementia *N* = 1035		Non-Dementia *N* = 38,181		Odds Ratio (95% CI)	
Variable	n	%	n	%	Crude	Adjusted
Sex						
Male	506	48.9	20,329	53.2	Ref.	Ref.
Female	529	51.1	17,852	46.8	1.19 (1.05–1.35) **	1.32 (1.16–1.50) ***
Age, years						
30–44	120	11.6	15,409	40.4	Ref.	Ref.
45–64	302	29.2	15,981	41.9	2.43 (1.96–3.00) ***	2.48 (2.00–3.08) ***
65+	613	59.2	6791	17.8	11.6 (9.51–14.1) ***	11.7 (9.41–14.6) ***
Income, NTD/month						
<19,200	395	38.2	12,433	32.6	1.66 (1.40–1.98) ***	1.32 (1.10–1.58) **
19,200–28,799	448	43.3	15,712	41.2	1.49 (1.26–1.77) ***	1.11 (0.93–1.34)
28,800+	192	18.6	10,036	26.3	Ref.	Ref.
Urbanization						
1 (highest)	263	25.4	10,268	26.9	1.01 (0.85–1.19)	1.08 (0.90–1.28)
2	294	28.4	11,466	30.0	1.01 (0.86–1.19)	1.11 (0.94–1.31)
3	304	29.4	11,952	31.3	Ref.	Ref.
4 (lowest)	174	16.8	4495	11.8	1.52 (1.26–1.84) ***	1.32 (1.09–1.61) **
Medical history						
Hypertension	507	49.0	10,944	28.7	2.39 (2.11–2.71) ***	0.91 (0.78–1.06)
Diabetes	244	23.6	5482	14.4	1.84 (1.59–2.13) ***	1.02 (0.87–1.20)
Hyperlipidemia	251	24.3	6867	18.0	1.46 (1.26–1.69) ***	0.98 (0.83–1.15)
Coronary heart disease	268	25.9	5033	13.2	2.30 (2.00–2.65) ***	1.05 (0.89–1.23)
Atrial fibrillation	22	2.13	478	1.25	1.71 (1.11–2.64) *	0.60 (0.38–0.94)
COPD	272	26.3	5602	14.7	2.07 (1.80–2.39) ***	0.93 (0.80–1.09)
Cancer	274	26.5	10,764	28.2	0.92 (0.80–1.06)	
Heart failure	87	8.41	1367	3.58	2.47 (1.97–3.10) ***	0.92 (0.72–1.18)
Depression	94	9.08	2928	7.67	1.20 (0.97–1.49)	
Liver disease	189	18.3	7615	19.9	0.90 (0.76–1.05)	
Chronic Infection	39	3.77	1029	2.70	1.41 (1.02–1.96) *	0.87 (0.62–1.23)
Neurologic clinic visit, n						
0	80	7.73	4806	12.6	Ref.	Ref.
1–5	151	14.6	11,440	30.0	0.79 (0.60–1.04)	1.16 (0.88–1.53)
6–20	307	29.7	10,806	28.3	1.71 (1.33–2.19) ***	2.43 (1.88–3.15) ***
>20	497	48.0	11,129	29.2	2.68 (2.11–3.41) ***	4.04 (3.15–5.18) ***

COPD, Chronic obstructive pulmonary disease; CI, confidence interval. * *p* < 0.05, ** *p* < 0.01, *** *p* < 0.001.

## Data Availability

The data that support the findings of this study are available from the insurance authority of Taiwan upon reasonable request after IRB approval.

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
