# Peer review of "Epileptic Patients with More Clinic Visits Are More Likely to Be Diagnosed with Dementia—A Population-Based Retrospective Cohort Study"

_diagnostics, 2024, doi:10.3390/diagnostics14232748_

Round 1

Reviewer 1 Report

Comments and Suggestions for Authors

It is an interesting paper in the sense that the increasing compliance and frequent hospital visits are associated with increasing incidence of dementia. The authors suggested that the frequent visits may mean more intractable epilepsy. The also pointed out that early development of dementia within one year may mean the bi-directionality between dementia and epilepsy.

The following question should be discussed.

The intractable epilepsy usually have polytherapy. Can your national data provide the data about the number of AEDs? And the number of admission due to epilepsy is also important.

If this data can be included in the paper, it would elucidate the intractability of the epilepsy patients. If your data cannot provide this data, please discuss it in the paper.

Author Response

[Response to the reviewer 1]

Response: Thank you for the kindly suggestion.

  1. The aim of this study to evaluate the benefit of the increasing compliance and frequent hospital visits to detect the other comorbidities, such as early dementia. According to the previous studies, those patients who suffered from epilepsy have the higher risk to develop dementia, the narrow ETGs can help doctors and patients to find dementia earlier and control it.

Inevitably, the intractable epilepsy is identified in this study, however, the epileptic attacks is not parallel to the frequency of followed-up. Poor compliance is one of causes of intractable epilepsy.

Response: Thank you for the inspirational comment on the relationship between the numbers of AEDs and epilepsy. However, due to the 2-year quota of using the database has been expired, we couldn’t look into this issue further. We will consider about this in further study in the future.

  1. Can your national data provide the data about the number of AEDs? And the number of admission due to epilepsy is also important.

Response: Thank you for the inspirational suggestion. Unfortunately, we are unable to conduct further data analysis. This is because the insurance authority gave us a 2-year term for using the database, we are now not allowed to use the database. It would take us additional efforts including time and funds to apply for a new database. Our apologies for being unable to conduct further data analysis now.

Reviewer 2 Report

Comments and Suggestions for Authors

1-)abbreviated version of epilepsy treatment gap can be removed from the keywords.

​2-)in the introduction,you can share research articles comparing cognitive skills of patients with and without epilepsy treatment. ​

3-)you should focus on one idea in one paragraph (especially introduction).

4-)you can write unabbreviated version of NHIRD.

5-)if some of the patients were excluded you can mention it in the methods section. how many patients were excluded?

6-)you can improve figure 1.

7-)you can use chatbots to improve figure 1.

8-)you can explain it further:

Both men and women had similar trends.

9-)figure 2 seems unclear too.

10-)you can discuss health system in Taiwan in the discussion section and its potential impact on study results should be discussed as well.

Comments on the Quality of English Language

the manuscript should be improved.

Author Response

1-)abbreviated version of epilepsy treatment gap can be removed from the keywords.

Response: Thank you for your precious suggestion.

We have removed the abbreviated version of epilepsy treatment gap from the keywords. Thank you!

​2-)in the introduction,you can share research articles comparing cognitive skills of patients with and without epilepsy treatment. ​

Response: Thank you for the valuable suggestion.

In the revision, we have cited a study by Zawar et al.: “Furthermore, in the recent longitudinal, multicenter cohort studies Zawar et al. found that both cognitively normal adults and those with mild cognitive impairment with poor-controlled/active seizures were at a near 2-fold higher risk to develop earlier cognitive decline. [17] [18]” (Please see page 2 paragraph 4 lines 4-8).

3-)you should focus on one idea in one paragraph (especially introduction).

Response: Thank you for the kind reminder.

We have had the manuscript revised avoiding long paragraphs.

4-)you can write unabbreviated version of NHIRD.

Response: Thank you for the suggestion.

We changed the NHIRD to unabbreviated version (National Health Insurance Research Database).

5-)if some of the patients were excluded you can mention it in the methods section. how many patients were excluded?

Response: Thank you for the kind reminder.

In the revision we have stated: “After excluding 281,343 patients who were ineligible for this study, 39,216 cases of epilepsy patients were included in the epilepsy cohort. The date with epilepsy diagnosed was defined as the index date. We used similar method to exclude 1,620,117 persons among individuals without history of epilepsy. From 8,917,940 persons without the history of epilepsy and/or dementia, the comparison cohort were selected matched by propensity scores with the sample size 4-fold of the epilepsy cohort “ (Please see page 3 lines 3-9.)

6-)you can improve figure 1.

Response: Thank you for the suggestion.

We advanced the image resolution unto 300 dpi, please see the new figure 1.

7-)you can use chatbots to improve figure 1.

Response: Thank you for the suggestion.

We will learn to use chatbots.

8-)you can explain it further:

Response: Thank you for the suggestion. We have further described method to established Control group:

“We used similar method to exclude 1,620,117 persons among individuals without history of epilepsy. From 8,917,940 persons without the history of epilepsy and/or dementia, the comparison cohort were selected matched by propensity scores with the sample size 4-fold of the epilepsy cohort.” (Pleas see 3 lines 5-8.)

9-) figure 2 seems unclear too.

Response: Thank you for the smart suggest.

We promoted the resolution unto 300 dpi, please see the new figure 2.

10-) you can discuss health system in Taiwan in the discussion section and its potential impact on study results should be discussed as well.

Response: Thank you for the inspirational reminder. We have slightly elaborated in the discussion:

“Known as National Health Insurance instituted in 1995 by merging all public health insurance program, this universal health care system has covered over 95% of population. This healthcare provides a high quality care comprehensive coverage, with low burden of health care costs. [23] However, we were surprised to find that only 9.3% of patients with epilepsy in our study were adequately adherent to their treatment.” (Please see page 18 paragraph 2 lines 1-5.).

The 2 paragraphs of statement below may also related to your comment:

“Our case-control analysis revealed that epilepsy patients living in the least urbanized rural areas were at an elevated risk of dementia. High-insurance coverage, easy-accessibility and low copayment of healthcare system in Taiwan make high clinical visit frequency [34]. Individuals living in area with lowest urbanization level have least chance to use medical services compared to those area in higher urbanization level. The medical divergence between urbanization levels are major factor associated with the possibility and frequency of medical service. [35-37] We suspect that epilepsy patients living in rural areas were likely to have less clinic visits. The elevated dementia risk in the rural areas was likely a truth, which might be associated with poor rural risk factors instead of treatment compliance. The recent studies reported that patients with poor compliance or fewer neurological consultations for seizure might have quicker decay in cognitive function based on the recent research. [17] [18]

Current health care institutions in Taiwan are divided into four levels: medical centers, reginal hospitals, district hospital and clinics. Obviously, large hospitals located in higher urbanization areas provide higher health-care-seeking choices and behaviors due to more diversity in medical care types with easy-accessibility and low medical cost burden. The “hierarchical medical system” fails to reduce unnecessary healthcare-seeking and medical resources wasting [35] [36].” (Please see page 19.)

11) Comments on the Quality of English Language the manuscript should be improved.

Response: Thank you for the reminder. An epidemiologist in the US has proofread the revised manuscript.